# Successful Establishment of Hepatitis E Virus Infection in Pregnant BALB/c Mice

**DOI:** 10.3390/v11050451

**Published:** 2019-05-17

**Authors:** Chenchen Yang, Xianhui Hao, Yunlong Li, Feiyan Long, Qiuxia He, Fen Huang, Wenhai Yu

**Affiliations:** 1Medical School, Kunming University of Science and Technology, Kunming 650500, China; yangchenchen0329@163.com (C.Y.); haoxianhui226@163.com (X.H.); liyunlong9018@163.com (Y.L.); longfeiyan24@163.com (F.L.); heqiuxia19941105@163.com (Q.H.); 2Institute of Medical Biology, Chinese Academy of Medical Sciences and Peking Union Medical College, Kunming 650118, China

**Keywords:** hepatitis E virus, pregnancy, miscarriage, vertical transmission

## Abstract

Worldwide, the Hepatitis E virus (HEV) is the main pathogen of acute viral hepatitis, with an extremely high mortality in pregnant women. However, the pathogenesis of HEV infection in pregnant women remains largely unknown. We established an HEV-infected pregnant mice animal model to explore the adverse pregnancy outcomes of HEV infection. Mice were infected with HEV in their early, middle and late stages of pregnancy. HEV RNA was detected in the tissues (liver, spleen, kidney, colon, uterus and placenta) of pregnant mice. HEV antigens were also detected in these tissues of HEV-infected pregnant mice. Miscarriages (7/8, 87.5%) occurred in pregnant mice infected with HEV in the middle of pregnancy. Th1-biased immune status was found in these aborted mice. Vertical transmission was confirmed by HEV replication in the uterus and placenta, as well as in the positive HEV RNA and HEV antigen positive in fetal livers. The successful establishment of HEV infection in pregnant mice is beneficial for further study of HEV pathogenesis, especially the adverse pregnancy outcomes caused by HEV infection.

## 1. Instruction

Hepatitis E virus (HEV) is an emerging infectious agent that causes acute viral hepatitis worldwide. Each year, more than 20 million estimated cases of HEV infection occur globally, resulting in 70,000 deaths [1]. HEV has at least eight genotypes [2,3]. Genotypes 1 and 2 are known to circulate in developing countries for more than half a century and only infect humans [1]. HEV genotypes 3, 4, and 7 can zoonotically transmit between animals and humans and are the ones that are considered emergent [2]. Hepatitis E is usually self-limited, with a case-fatality rate of <1% among the general population [3], but a significantly high mortality (20%~30%) in the third trimester of pregnant women [4,5,6]. Genotype 1 HEV infection during pregnancy has been reported to result in severe placental diseases, such as hemorrhage, miscarriage, and stillbirth [7,8,9]. Genotype 4 HEV infections in pregnant women have usually been ignored. Recently, 33.33% (8/24) of anti-HEV IgM positive pregnant women in China where the prevalent genotype 4 HEV has adverse pregnancy outcomes, including four threatened preterm births, three premature ruptures of membranes and one threatened abortion [10]. More severe pregnancy outcomes were reported in both anti-HEV IgM and IgG positive pregnant women with a 72.22% (13/18) occurrence of poor pregnancy outcomes, including 10 threatened preterm births, two premature ruptures of membranes and one threatened abortion [11]. Genotype 4 HEV infection in pregnant women in China resulted in stillbirth, preterm premature rupture of membranes, and fetal distress; and pregnancy induced hypertension syndrome has also been reported [11,12]. However, the pathogenesis of HEV infection in pregnant women remains unknown.

The interplay of hormonal and immunologic changes during pregnancy, along with the high viral load of HEV, may be the reason for woman being more vulnerable to HEV [13,14]. HEV replication in placental-derived cells (JEG-3) [15] and in a maternal-fetal interface using the decidua basalis and fetal placenta [7] in vitro made great contributions to the study of HEV pathogenic mechanisms. However, an animal model is necessary to investigate the underlying severity of hepatitis E in pregnancy and to better understand its pathogenesis. HEV-infected pregnant rabbit animal models have been successfully established, resulting in a 75% mortality rate and a 33.3% miscarriage rate [16]. Vertical transmission was confirmed by the presence of HEV RNA and antigen in the placenta of HEV-infected rabbits [16]. Although rabbit HEV can infect non-human primates [17] and the recently isolated HEV from humans shares a close relationship with rabbit HEV [18], the rabbit HEV shares less than an 82% similarity to human HEV or swine HEV [19]. Therefore, an alternative or common animal model, for example, one using BALB/c mice, which are sensitive to HEV infection [20,21], should be established to further study the mechanism of HEV infection in pregnancy.

In the present study, pregnant BALB/c mice were successfully infected with genotype 4 swine HEV. HEV RNA was detected in the feces, serum, and tissues by RT-nested PCR (RT-nPCR) and quantitative RT-PCR (qRT-PCR). Meanwhile, HEV antigens were analyzed in the liver, spleen, kidney, and colon by immunohistochemistry. Moreover, vertical transmission was confirmed by the presence of HEV RNA and HEV antigens in the uterus, placenta, and fetal livers of the newborn mice.

## 2. Materials and Methods

### 2.1. Ethics Statement

All animal experiments were approved by the Animal Care and Use Committee of Kunming University of Science and Technology. We followed the guidelines of the Kunming University of Science and Technology Animal Care and Use Committee when handling the experimental animals during this study. Thirty-six female BALB/c mice, 6 week old, weighing 18~22 g, were obtained from SLAC Animal Laboratory (China). Their food and water were sterilized, and the experimental animal rooms were fumigated. Prior to inoculation with HEV, the mice were confirmed negative for HEV antibodies by enzyme-linked immunosorbent assay (ELISA), and no HEV RNA was detected in the serum and feces of all experimental animals.

### 2.2. Virus

Swine fecal samples containing HEV genotype 4 (KM01, GenBank No. KJ155502) were obtained from a village in Southwestern China. The fecal suspension (10%) was centrifuged at 12,000 × *g* at 4 °C for 10 min, filtered through 0.45 and 0.22 µm microfilters, and treated with penicillin and streptomycin at 4 °C for 1 h. The HEV titer in suspension was 1.0 × 10^4^ copies/mL, and the fecal suspension was stored in liquid nitrogen until use.

### 2.3. Experimental Design

All female mice (*n* = 36), as candidates for pregnancy, were randomly divided into four groups as illustrated in Figure 1. Normal copulated group (MOCK, *n* = 9), mice were injected with PBS after copulation; the early pregnancy of the HEV infection group (*n* = 9) occurred when individual female mice were copulated with individual adult male mice for 3 days. Then the mice were intravenously (i.v.) injected with HEV (200 μL); during the middle pregnancy of the HEV infection group (*n* = 9), individual female mice were pre-copulated with individual adult male mice, then HEV was intravenously (i.v.) inoculated (200 μL) after 10.5 days of copulation; during the late pregnancy of the HEV infection group (*n* = 9), individual female mice were pre-copulated with individual adult male mice. Then HEV was intravenously (i.v.) inoculated (200 μL) after 14 days of copulation. The insemination process occurred as follows: One female mouse copulated with one healthy adult male mouse (8 to 10 weeks old) in individual cage, then the couple was separated when the vaginal plug was observed 1~3 days after being caged together. To determine whether the mouse was pregnant, estrogen and progesterone concentration were tested. At 7, 14 and 19/20/21 days post-copulation, the serum estrogen and progesterone levels were detected. If the level of estrogen and/or progesterone kept elevating, the mice were defined as pregnant; if the level elevated first and then declined, the pregnancy was defined as a miscarriage; if it remained unchanged, the mice were defined as non-pregnant.

Non-pregnant mice in the early pregnancy group (*n* = 4) at 7 dpi were treated with ribavirin (RBV, 50mg/kg/day) for 10 days to evaluate the specificity of the HEV titer. All mice were humanely euthanized and necropsied after delivery, and had a cesarean section or treatment at 19–21 days post-copulation. Stool and blood were collected for HEV RNA detection. Tissues, including liver, spleen, kidney, colon, uterus, placenta, and fetus liver (if available) were collected and stored at −80°C for HEV RNA and antigen detection or fixed in a 10% neutral buffered formalin for histopathological and/or immunohistochemistry analysis.

### 2.4. Determination of Pregnancy-Related Hormones

To confirm the pregnancy, serum estrogen and progesterone in female mice were determined by EIA Kit following the manufacturer’s instructions (Cayman Chemical, Ann Arbor, Michigan CA, USA).

### 2.5. HEV RNA Detection and Quantification

Total RNA was extracted from all samples by Trizol (Invitrogen, Carlsbadd, Califormia America), according to the instructions of the manufacturer. Reverse transcription was performed using M-MLV Reverse Transcriptase (Takara, Kusatsu, Japan) according to the directions of the manufacturer. A 348 nt amplicon from HEV ORF2 was amplified by RT-nPCR as described previously [22]. Negative strand RNA of HEV was detected to further confirm replication according previous study [23]. HEV titer was quantified by SYBR green-based qRT-PCR with HEV-specific primers as previous studies [24,25].

### 2.6. Detection of Cytokine Levels

The levels of Th1 cytokine (interleukin 12, IL-12) and Th2 cytokine (interleukin 6, IL-6) in serum samples were assayed using commercial enzyme-linked immunosorbent assay (ELISA) kits (4A Biotech Beijing, China) according to the manufacturer’s instructions.

### 2.7. Immunohistochemistry

For immunohistochemistry (IHC), the tissues were deparaffinized, hydrated, water bath-heated for antigen retrieval, and then blocked with the addition of 3% hydrogen peroxide for 10 min. Tissue sections were incubated overnight at 4 °C with HEV ORF2 specific antibodies (Millipore, Temecula, CA, USA, MAB8003, 1:250 dilutions), washed with PBS for three times, and then incubated with an HRP-labeled goat anti-Mouse IgG antibody at 37 °C for 60 min. After being washed three times, a DAB substrate (Abcam, Cambridge Science Park, England) was added and Gill’s hematoxylin was applied as a background stain. The slides were sealed with neutral balsam, inspected, and photographed under a microscope.

### 2.8. Histopathology

Tissues for histologic examination were fixed in a 10% neutral buffered formalin, routinely processed, sectioned at a thickness of 3 μm, and then stained with hematoxylin and eosin (H & E). The samples were photographed and analyzed under a microscope (Nikon, Tokyo, Japan) equipped with a digital camera.

### 2.9. Statistical Analysis

All experiments were performed at least thrice. GraphPad Prism 5.01 software was used for statistical analysis. A student-*t* test analysis was used to determine the significance of differences between two groups. *p* value < 0.05 was considered statistically significant.

## 3. Results

### 3.1. HEV Successfully Infected Pregnant BALB/c Mice

Hepatitis E infection causes a high mortality (about 20%) in pregnant women, with an increasingly high incidence and severity in the third trimester of pregnancy [26]. However, the mechanisms for this high morbidity and mortality during pregnancy remains unclear, because of the unavailable animal models. In this study, pregnant mice in different trimesters of pregnancy were inoculated with HEV. To simulate HEV-infection in pregnant women in different trimesters of pregnancy, mice were inoculated with HEV at 3 days (early pregnancy), 10.5 days (middle pregnancy), and 14 days (late pregnancy) post-copulation. Four mice in the early pregnancy group were not pregnant, and one mouse in the middle group and one mouse in the late pregnancy were not pregnant, as defined by unchanged estrogen and/or progesterone (Figure 1, Figure 2A,B). Interestingly, HEV RNA was detected in the feces at 3~5 dpi and in the serum at 5~7 dpi (Figure 2C–E, Table 1). Viremia was observed in all infected pregnant mice (Figure 2). The viral titer in the serum of the HEV-infected miscarriage mice was compared with that of the non-pregnant mice and delivery mice at 7 dpi in the early and middle groups (Figure 2F). A negative strand of HEV RNA was also detected in the serum (Table 1). The relative expression of negative-strand HEV RNA in the serum was compared among non-pregnant, normal delivery and miscarriage mice at 7 dpi in the early and middle pregnancy groups (Figure 2G). To evaluate the specificity of HEV, RBV was applied to treat HEV-infected non-pregnant mice in the early pregnancy group. Fortunately, mild antiviral effects were obtained in the liver and uterus of mice treated with RBV compared with the untreated mice (Figure 2H). In addition, positive and negative strands of HEV RNA were also detected in the liver, spleen, kidney, and colon of HEV-infected pregnant mice (Figure 2I–K, Table 1). All pregnant mice infected with HEV in the early pregnancy (5/5) or the late pregnancy (8/8) has a normal delivery, while only one mouse infected with HEV in the middle pregnancy had a normal delivery (1/8). Miscarriages (7/8, 87.5%) happened in the mice infected with HEV in the middle of pregnancy (Figure 1 and Figure 2). The feces, serum, and tissues of mock mice inoculated with PBS were all negative to HEV RNA.

To further identify the replication of HEV in mice, HEV antigens were detected by IHC. Consistent with HEV RNA in tissues of mice, HEV antigens were distributed in the liver, spleen, kidney, and colon of pregnant mice, including normal delivery and aborted mice (Figure 3). However, the uninfected mock mice were negative to the HEV antigen. The results strongly indicate that HEV can replicate in pregnant mice.

### 3.2. HEV Infection Leads to Adverse Pregnancy Outcome

HEV infection usually causes self-limiting diseases but leads to high maternal mortality in pregnant women. The pathogenesis of HEV infection in pregnant women is largely unknown because of the unavailable animal models. Thus, these successfully established pregnant BALB/c mice models were used to assess the adverse effects of HEV infection. Although there is no maternal death, seven mice infected with HEV (7/8, 87.5%) in the middle pregnancy had a miscarriage (Figure 1 and Figure 4A). Notably, the number of fetuses from HEV-infected mothers was significantly less than that from the uninfected mock mothers (a total of 111 fetuses in the mock group vs the HEV-infected group in early (35 fetuses), middle (3 fetuses), or late (61 fetuses) pregnancy (Figure 4B).

Normal pregnancy is associated with a Th2 biased peripheral cytokine profile [27]. It was reported that Th1 biased immune status is abortion-prone [28]. The fine modulate Th1/Th2 balance is a critical factor in the protection of the fetus against abortions in mice [29]. The Th1 cytokines such as IL-12 and INF-γ can cause fetal loss, whereas the Th2 cytokines, such as IL-6 and IL-10 are protective [30]. To explore the association between HEV infection and abortion, IL-12 (Th1 cytokine) and IL-6 (Th2 cytokine) were determined by ELISA at 19–21 days post-copulation (Figure 4C,D). Uninfected mice with normal delivery showed a Th2 biased immune status, while a higher IL12/IL6 ratio (9.48 folds) was found in the HEV-infected miscarriage mice (Figure 4E). Th1 biased immune status was observed in these aborted mice (Figure 4E). Thus, the disbalance of the Th1/Th2 immune status in HEV-infected mice may contribute to the high rate of miscarriage.

### 3.3. HEV Vertically Transmits from Mother to Fetus

HEV can vertical transmit to the fetus, but the mechanism is remaining elusive. The uterus supplies a safe place for the fetus to develop [31]. The placenta forms the interface between the mother and the fetus, and is able to maintain two separate circulatory systems [32]. The blood-placenta barrier (BPB) is an important barrier to protect the fetus against pathogen infection. However, whether HEV can replicate in the uterus and/or placenta is unclear.

In this study, positive and negative strands of HEV RNA were detected in the uterus and placenta of all the HEV-infected mothers, no matter whether they were infected with HEV during early, middle, or late pregnancy; however, one placenta of a mouse infected in late pregnancy was below the cutoff value (Figure 5A). In addition, HEV antigens were observed in the uterus of all HEV-infected mice, those with both normal deliveries and aborted pregnancies (Figure 5B). Meanwhile, the placenta was also positive to HEV antigens (Figure 5B). Moreover, histopathological changes were analyzed in the uterus and placenta of HEV-infected mice. Severe inflammation responses with inflammatory exudates and hemorrhages were observed in the uterus of aborted mice infected with HEV in middle pregnancy (Figure 5C). Results indicate that HEV replicates in the uterus and placenta.

We investigated whether the fetus suffers from HEV infection, since the uterus and placenta of mother were infected by HEV. Thus, all the livers of fetuses born from HEV-infected mothers who were infected with HEV at the early (fetus = 35), middle (fetus = 3) or late (fetus = 61) stages of pregnancy were used to evaluate HEV infection. Surprisingly, all the livers of fetuses born from HEV-infected mothers were positive for HEV RNA (both positive and negative strands of HEV RNA), except two fetuses that were negative for HEV, whose viral titers were below the cut-off value (Figure 5A). Furthermore, HEV-positive stains were observed in the livers of fetuses born from HEV-infected mothers (Figure 5B). Results strongly suggest that HEV can replicate in the uterus and placenta and vertically transmit to fetuses.

## 4. Discussion

HEV infection is the leading cause of acute viral hepatitis [1,33], which causes significant morbidity and mortality during pregnancy [34,35]. Miscarriage and stillbirth were reported in pregnant women infected by genotype 1 and genotype 4 HEV [10,36]. Genotype 3 HEV infection was thought to be less-pathogenic in the maternal-fetal interface [7]. Genotype 4 HEV infection is more severe than genotype 3 HEV infection [37]. Li et al., reported the adverse pregnancy outcomes, including preterm birth, premature rupture of membranes and abortion, caused by genotype 4 HEV infection in China [10], which indicated that a genotype 4 HEV infection during pregnancy could also lead to severe adverse outcomes. However, the mechanism of high morbidity and mortality during pregnancy remains unclear.

The extremely high mortality in HEV-infected pregnant women urgently prompted us to establish an animal model of HEV infection in pregnancy to further study the pathogenesis. Although human HEV-infected non-human primates or swine fail to simulate the adverse outcomes of HEV infection in pregnancy [38,39], pregnant rabbits were successfully infected with HEV, which caused a 75% mortality rate and 33.33% miscarriage rate [16]. Murine models, such as the BALB/c nude mouse [20], Mongolian gerbils [40] and BALB/c mouse [21] have been reported to be susceptive to genotype 4 HEV infection, which is isolated from China. It has been reported that genotype 1 and 3 HEV can infect humanized mice with chimeric human livers, but fail to infect C57BL/6 mice [41,42,43]. Thus, to further identify whether genotype 4 swine HEV will lead to high mortality and adverse outcomes in BALB/c mice, the most common animal model of HEV-infected pregnant mice was performed. It is encouraging that a high miscarriage rate (87.5%) was observed in pregnant mice when infected with HEV in middle pregnancy. In the early pregnancy group, four mice were non-pregnant with unchanged estrogen and progesterone levels at 7 days post-copulation. It is possible that the miscarriage happened earlier than 7 days after pregnancy. Although higher viral titers were observed in the miscarriage mice compared with non-pregnant or normal delivery mice at 7 dpi, the different time points of post-copulation in the early and middle pregnancy groups may affect this result.

Moreover, a higher IL-12/IL-6 ratio (Th1-biased state) was found in HEV-infected mice with miscarriages than in uninfected MOCK or HEV-infected mice with normal deliveries. These HEV-infected mice with normal deliveries showed Th2-biased states, which are important to maintain pregnancy. The altered immune state caused by HEV infection during pregnancy from Th2-biased to Th1-biased has been observed in HEV-infected pregnant women with fulminant hepatic failure [27], which may be a miscarriage marker and should be carefully observed. However, no maternal death occurred in the HEV-infected mice, possibly because of the mice’s short pregnancy cycle (21 days), the short-time inoculation of HEV (only 17 dpi, 11 dpi, and 5 dpi) or the moderate clinical symptom caused by genotype 4 HEV. The pregnant BALB/c mice model is useful to study HEV infection during pregnancy because BALB/c mice are readily available, low-cost and easy to handle.

Vertical transmission of HEV infection has been noted since 1995 [44]; this type of transmission threatens fetal and neonatal health, and more importantly, results in maternal death. Miscarriage, stillbirth, or neonatal death were observed in 56% of infants whose mothers were infected with HEV [45]. However, the vertical transmission of HEV was controversial, because of the transmission rate from mother to infant varied from 100% [46] to 17% [47]. In addition, only a few studies identified that HEV replicates in placental tissues from HEV-infected women or in the maternal-fetal interface [7,48]. In the present study, HEV RNA and a large portion of HEV antigens were detected in the uterus and placenta of all HEV-infected mice, indicating that HEV replicates in the uterus and the placenta, which may contribute to miscarriage. Furthermore, we found that HEV can cross the blood placental barrier to infect the fetus. In addition, all the fetuses were infected by their HEV-infected mother, and the liver of an HEV-infected fetus was positive for HEV RNA and antigens. The transmission of HEV through the fecal-oral route from mother to their newborn fetus can be ignored because all baby mice gave birth by cesarean section. Although the adverse pregnancy outcomes of genotype 4 HEV infection in pregnant women have usually been ignored, preterm births, premature rupture of membranes, and abortions were recently reported in clinical [10,11,12]. In the present study, a high miscarriage rate was found in pregnant mice infected by HEV in the middle stage of pregnancy, with severe inflammation responses in the uterus. Thus, more attention must be paid to pregnant women with jaundice, or an elevated activity/level of liver enzymes. There are no specific anti-HEV drugs in clinical, but RBV as an off-label drug is usually used for chronic hepatitis E treatment. In the present study, almost no antiviral effect was obtained after RBV treatment, because the KM01 strain contains a G1634R mutation in ORF1, which is reportedly associated with the treatment failure of RBV therapy in acute and chronic hepatitis E patients [49]. More importantly, the teratogenicity limits its use during pregnancy.

In summary, HEV infection in pregnant mice was successfully established with a high rate of miscarriage. This animal model can be used to model HEV infection in pregnant women to study the adverse outcomes caused by HEV and to explore host response and pathogenesis. This model is also important for future anti-HEV drug development.

## Figures and Tables

**Figure 1 viruses-11-00451-f001:**
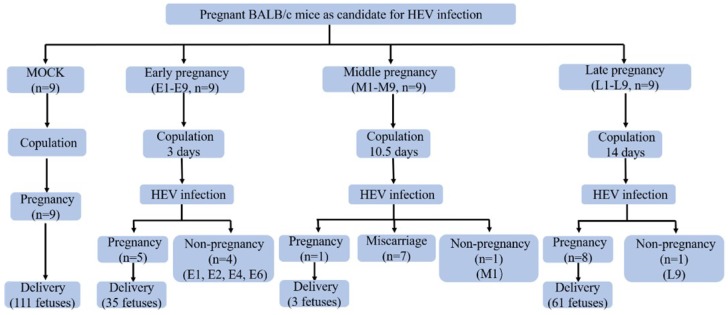
Effects of Hepatitis E virus (HEV) infection in the pregnant BALB/c mice.

**Figure 2 viruses-11-00451-f002:**
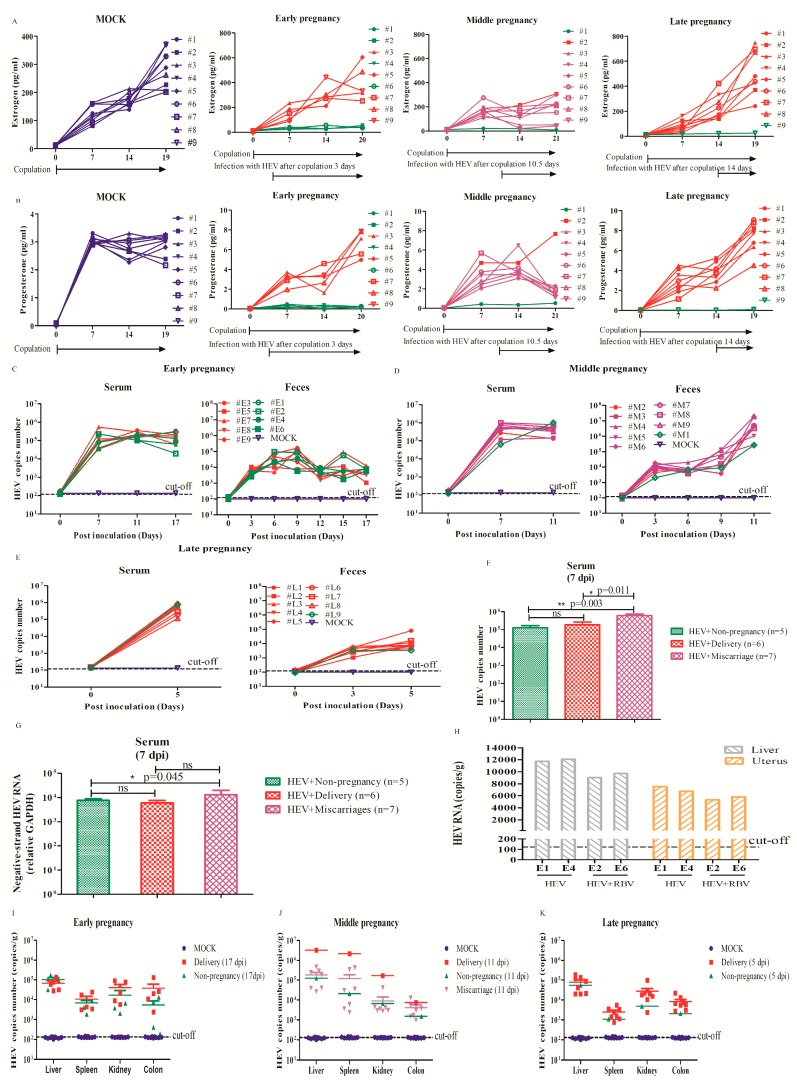
Profile of HEV infection in pregnant mice. The concentration of estrogen (**A**) and progesterone (**B**) in mice infected with or without HEV in the early, middle and late stages of pregnancy. The viral titer of HEV in the feces and serum of mice in early (**C**), middle (**D**) and late (**E**) pregnancy. (**F**) Comparison of the viral titer in serum among non-pregnant (*n* = 5), delivery (*n* = 6) and miscarriage (*n* = 7) mice in early and middle pregnancy at 7 dpi. (**G**) The relative expression of negative-strand HEV RNA in the serum of HEV-infected non-pregnant (*n* = 5), delivery (*n* = 6) and miscarriage (*n* = 7) mice in early and middle pregnancy at 7 dpi. (**H**) The viral titer of HEV in liver and uterus in HEV-infected non-pregnant mice in early pregnancy treated with or without or RBV. (**I–K**) The viral titer of HEV in the tissues of mice in early (**I**), middle (**J**) and late (**K**) pregnancy. Mock mice are drawn in a blue colour, non-pregnant mice are drawn in a green colour, delivery mice are drawn in a red colour, and miscarriage mice are drawn in a purple colour.

**Figure 3 viruses-11-00451-f003:**
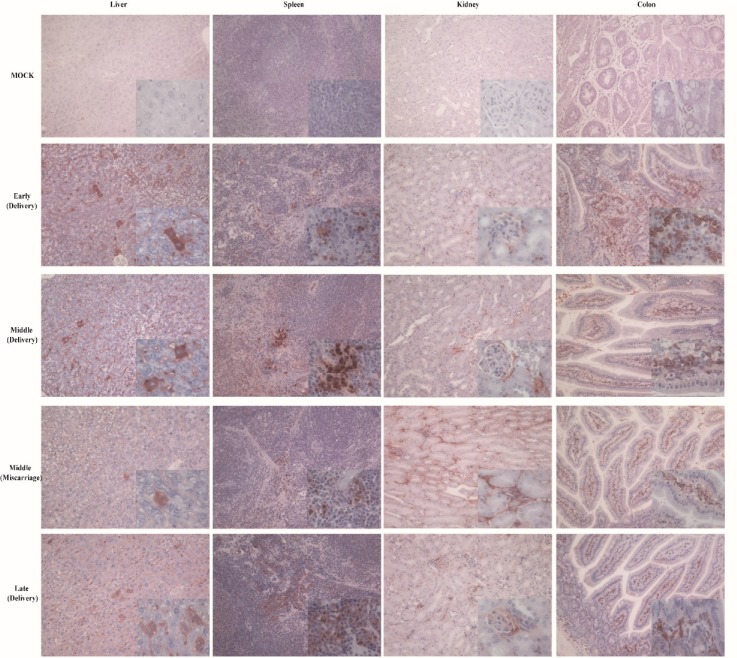
HEV antigens detected by IHC. HEV antigens detected in the liver, spleen, kidney and colon of mice infected with or without HEV, × 20.

**Figure 4 viruses-11-00451-f004:**
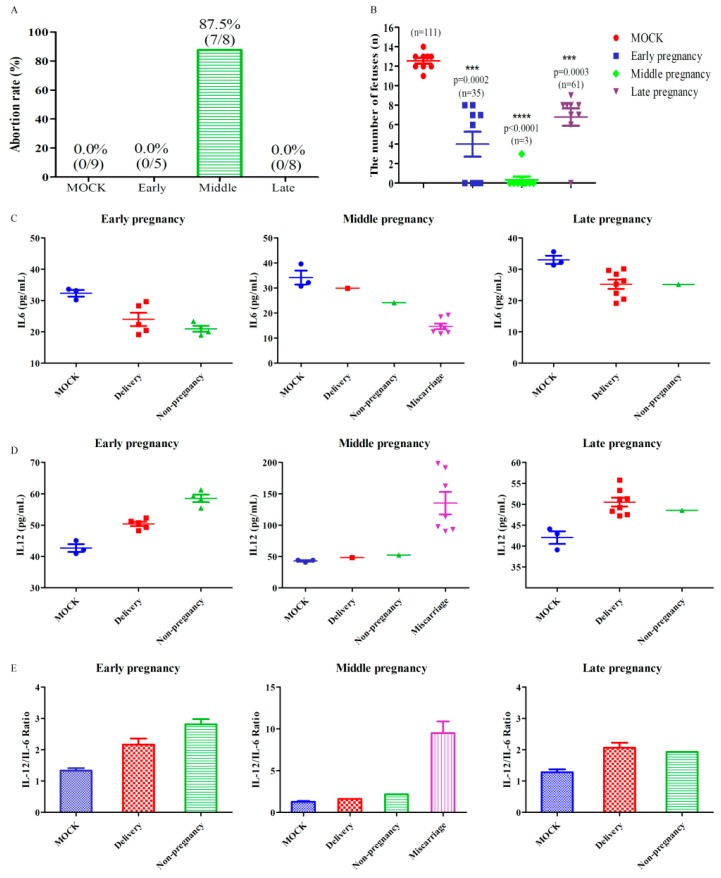
HEV infection leads to adverse pregnancy outcomes. (**A**) The rate of abortion. (**B**) The number of fetuses from an uninfected or HEV-infected mother. The concentration of IL-6 (**C**) and IL-12 (**D**) in the serum of mice with or without HEV infection. (**E**) The IL-12/IL-6 ratio in pregnant mice with or without HEV infection.

**Figure 5 viruses-11-00451-f005:**
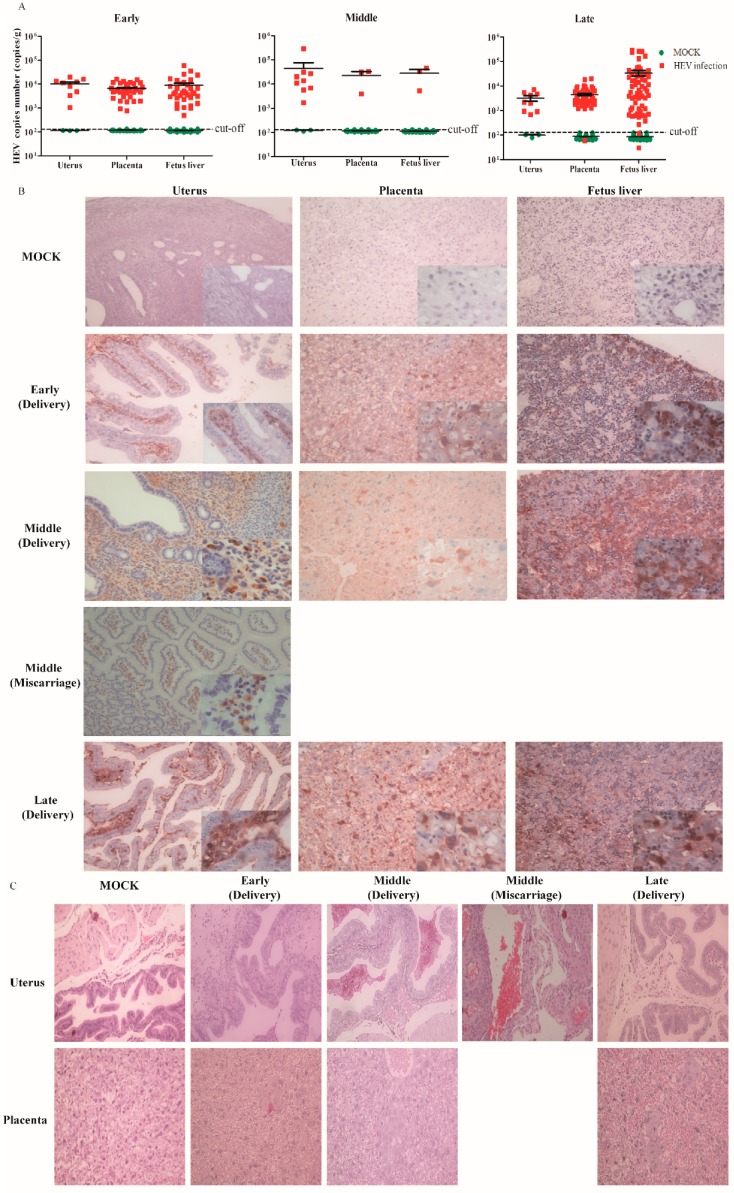
HEV vertically transmits from the mother to the fetus. (**A**) Viral titer in the uterus and placenta of HEV-infected pregnant mice or liver of fetuses. (**B**) HEV antigen analysis of the uterus, placenta and liver by IHC, × 20. (**C**) Histopathological analysis of the uterus and placenta of HEV-infected mice, × 20.

**Table 1 viruses-11-00451-t001:** HEV RNA detection in HEV-infected pregnant mice.

Sample	Strands	Groups									
		MOCK	Early Pregnancy			Middle Pregnancy			Late Pregnancy		
			Del	NP	Mis	Del	NP	Mis	Del	NP	Mis
Feces	Positive	0/9	5/5	4/4	0/0	1/1	1/1	7/7	8/8	1/1	0/0
	Negative	0/9	5/5	4/4	0/0	1/1	1/1	7/7	8/8	1/1	0/0
serum	Positive	0/9	5/5	4/4	0/0	1/1	1/1	7/7	8/8	1/1	0/0
	Negative	0/9	5/5	4/4	0/0	1/1	1/1	7/7	8/8	1/1	0/0
Liver	Positive	0/9	5/5	4/4	0/0	1/1	1/1	7/7	8/8	1/1	0/0
	Negative	0/9	5/5	4/4	0/0	1/1	1/1	7/7	8/8	1/1	0/0
Spleen	Positive	0/9	5/5	4/4	0/0	1/1	1/1	7/7	8/8	1/1	0/0
	Negative	0/9	5/5	4/4	0/0	1/1	1/1	7/7	8/8	1/1	0/0
Kidney	Positive	0/9	5/5	4/4	0/0	1/1	1/1	7/7	8/8	1/1	0/0
	Negative	0/9	5/5	4/4	0/0	1/1	1/1	7/7	8/8	1/1	0/0
Colon	Positive	0/9	5/5	4/4	0/0	1/1	1/1	7/7	8/8	1/1	0/0
	Negative	0/9	5/5	4/4	0/0	1/1	1/1	7/7	8/8	1/1	0/0
Uterus	Positive	0/9	5/5	4/4	0/0	1/1	1/1	7/7	8/8	1/1	0/0
	Negative	0/9	5/5	4/4	0/0	1/1	1/1	7/7	8/8	1/1	0/0
Placenta	Positive	0/111	35/35	0/0	0/0	3/3	0/0	0/0	60/61	0/0	0/0
	Negative	0/111	35/35	0/0	0/0	3/3	0/0	0/0	60/61	0/0	0/0
Fetus liver	Positive	0/111	35/35	0/0	0/0	3/3	0/0	0/0	59/61	0/0	0/0
	Negative	0/111	35/35	0/0	0/0	3/3	0/0	0/0	59/61	0/0	0/0

NP: non-pregnancy; Del: Delivery; Mis: Miscarriage.

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
