# Peer review of "Successful Establishment of Hepatitis E Virus Infection in Pregnant BALB/c Mice"

_viruses, 2019, doi:10.3390/v11050451_

Round 1
Reviewer 1 Report
Authors have adequately adressed the concerns
Author Response
Reviewer 1
Comments and Suggestions for Authors
Authors have adequately addressed the concerns
Answer: thank you for your help.
Reviewer 2 Report
The authors did do some of the requested changes, which helped improving the manuscript. However, there are still major issues that have not been answered and not addressed or forgotten. Please find the details comments:
Question 1:
The authors were requested to discuss the fact that they were able to infect mice with HEV, while different groups have not been. They state that they succeed to infect BALB/c mice, but not C57BL/6. This has to be mentioned in the manuscript.
Question 3:
The authors did not included the references in the literature although they state that this has been done. Schlosser to the section line 230-234 and the others as mentioned before.
Q4:
Grammar and spelling are still not on an appropriate level.
Q5:
It was requested to include RBV controls. Not for an evaluation as medication, but as a replication control to demonstrate specificity of HEV titers.
The data submitted in the response letter should be included in the paper, except for the Sofosbuvir samples. It must be discussed that none of the therapeutic interventions has an effect in the paper, since that is a surprising finding! This finding has to be discussed in the discussion section, because it would have implications for mouse models in general.
Q9:
Please quantify the levels of negative strand RNA. The table has to be labelled properly.
Q11:
Please explain the things shown in the picture submitted as a response to question 11. It is not understandable what there is to see.
Q16:
Include an RBV cohort for the
Author Response
Reviewer 2
Comments and Suggestions for Authors
The authors did do some of the requested changes, which helped improving the manuscript. However, there are still major issues that have not been answered and not addressed or forgotten. Please find the details comments:
Question 1:
The authors were requested to discuss the fact that they were able to infect mice with HEV, while different groups have not been. They state that they succeed to infect BALB/c mice, but not C57BL/6. This has to be mentioned in the manuscript.
Answer:The successful establishment of HEV-infected BALB/c mice model has been submitted and under review in BMC Infectious Diseases, we will supply to reviewer. We added in discussion section in the new version.
Question 3:
The authors did not included the references in the literature although they state that this has been done. Schlosser to the section line 230-234 and the others as mentioned before.
Answer:We added these references in the new version.
Q4:
Grammar and spelling are still not on an appropriate level.
Answer:We carefully edited in the new version.
Q5:
It was requested to include RBV controls. Not for an evaluation as medication, but as a replication control to demonstrate specificity of HEV titers.
The data submitted in the response letter should be included in the paper, except for the Sofosbuvir samples. It must be discussed that none of the therapeutic interventions has an effect in the paper, since that is a surprising finding! This finding has to be discussed in the discussion section, because it would have implications for mouse models in general.
Answer:We added RBV treatment in fig.2 in the new version.
Q9:
Please quantify the levels of negative strand RNA. The table has to be labelled properly.
Answer:We added the relative expression of negative strand HEV RNA in Fig.2 in new version, since there is no reported about the quantitation of negative strand HEV RNA. The table was modified in the new version.
Q11:
Please explain the things shown in the picture submitted as a response to question 11. It is not understandable what there is to see.
Answer:The reviewer ask about the less fetuses in HEV-infected group which was infected after implantation of eggs into placenta. We answered that the stillbirth was observed in the placenta (A) of HEV-infected mice shown in the followed figure, stillbirth or resorption fetus (B) were circled.
| A |
| B |
Q16:
Include an RBV cohort for the
Answer:same answer to Q5

Reviewer 3 Report
In the present revised manuscript, Yang et al have developed a model of HEV infection in pregnant BALB/c mice. The authors have answered most of the points raised during the first submission and improved the quality of their manuscript. However, a few points still need to be revised. The paper still needs additional English proofreading and some sentences need to be rephrased.
Overall, the development of a model of HEV infection in pregnant BALB/c mice is of particular interest for the field but the present manuscript still requires some improvement to be suitable for publication in Viruses.
- Line 36-39 and 221-224: In the paper cited (10), HEV-4a was detected in only one pregnant woman with adverse pregnancy outcome. It is likely that the other pregnant women were also infected with the same genotype, as it is prevalent in China. However, this should be better explained in the text.
- Line 46-48: References are lacking on rabbit HEV infection in non-human primate and the HEV strains related to rabbit HEV isolated in humans.
- The legend of table 1 is missing.
- Figure 2A and 2B: The estrogen and progesterone data for the mock group should be included in a separate graph. It could be useful to use to same colour code for all the graphs in Figure 2 (blue for all mock even in Figure 2A and 2B and green for non-pregnant mice) and better explain it in the figure legend.
- Figure 2F: Have you included the data from the 3 groups? What is the number of mice included for each group? Can you really compile and statistically compare these data as they come from 3 different groups infected at different times post-copulation and as for some of the conditions the number of mice is low?
- Line 149-150: It is difficult to compare and make conclusions when there is only 1 mouse for some of the groups.
- Line 176-177 and 241-242: Is normal pregnancy also associated with a Th2 biased peripheral cytokine profile in mice? Has this been checked in mice?
- Figure 4C-D: Why are the results presented as folds (normalised with mock) rather than as concentration (pg/ml) if the quantification was performed by ELISA. At which time point post-infection and post-copulation was this assay performed? It could be useful to also include a graph showing the IL12/IL6 ration of mock, delivery, non-pregnancy and miscarriage mice for each group (mock, early pregnancy, middle pregnancy and late pregnancy) as performed in Fig. 2G-I.
Author Response
Reviewer 3
Comments and Suggestions for Authors
In the present revised manuscript, Yang et al have developed a model of HEV infection in pregnant BALB/c mice. The authors have answered most of the points raised during the first submission and improved the quality of their manuscript. However, a few points still need to be revised. The paper still needs additional English proofreading and some sentences need to be rephrased.
Overall, the development of a model of HEV infection in pregnant BALB/c mice is of particular interest for the field but the present manuscript still requires some improvement to be suitable for publication in Viruses.
Q1: Line 36-39 and 221-224: In the paper cited (10), HEV-4a was detected in only one pregnant woman with adverse pregnancy outcome. It is likely that the other pregnant women were also infected with the same genotype, as it is prevalent in China. However, this should be better explained in the text.
Answer:Thank you. Less attention focus on HEV in China maybe the main reason why only one pregnant woman with adverse pregnancy outcome has been reported. In fact, HEV infection induced adverse pregnancy outcome are common in pregnant women in China, but most of the reports were in Chinese. For example, Zhong et al., reported a 10.26%-23.08% adverse pregnancy outcome in HE-infected pregnant women (n=78) in China from 2013-2015. Zhang et al., reported stillbirth, preterm premature rupture of membranes, fetal distress and pregnancy induced hypertension syndrome in HEV-infected pregnant women in Huaian, Jiangsu, China.
Meanwhile, there were not only one pregnant women with adverse pregnancy outcome in the reference 10. They reported 33.33% (8/24) adverse pregnancy outcomes in anti-HEV IgM positive pregnant women in China, including 4 threatened preterm births, 3 premature rupture of membranes and 1 threatened abortions. 72.22% (13/18) poor pregnancy outcomes in both anti-HEV IgM and IgG positive pregnant women, including 10 threatened preterm births, 2 premature rupture of membranes and 1 threatened abortions. We added these adverse pregnancy outcomes in the new version.
Q2: Line 46-48: References are lacking on rabbit HEV infection in non-human primate and the HEV strains related to rabbit HEV isolated in humans.
Answer:We added the references about the transmission of rabbit HEV in the new version.
Q3: The legend of table 1 is missing.
Answer:We added the legend of table 1
Q4: Figure 2A and 2B: The estrogen and progesterone data for the mock group should be included in a separate graph. It could be useful to use to same colour code for all the graphs in Figure 2 (blue for all mock even in Figure 2A and 2B and green for non-pregnant mice) and better explain it in the figure legend.
Answer:We changed as you suggested in figure 2.
Q5: Figure 2F: Have you included the data from the 3 groups? What is the number of mice included for each group? Can you really compile and statistically compare these data as they come from 3 different groups infected at different times post-copulation and as for some of the conditions the number of mice is low?
Answer:Fig.2F only show the viral titer of HEV in early and middle pregnancy groups at 7dpi (the same time point after infection). Because the mice in late pregnancy group were euthanized at 5 dpi, thus we didn’t compared it with early and middle group. The number of mice were shown in the new version.
Q6: Line 149-150: It is difficult to compare and make conclusions when there is only 1 mouse for some of the groups.
Answer:the comparison was performed in early and middle pregnancy groups, including non-pregnant (n=5), delivery (n=6), and miscarriage (n=7). We clarified in the figure 2 in the new version.
Q7: Line 176-177 and 241-242: Is normal pregnancy also associated with a Th2 biased peripheral cytokine profile in mice? Has this been checked in mice?
Answer:Yes, the balance of Th1/Th2 cytokines is also very important in mouse. Imbalance of Th1/Th2 cytokines is well known in spontaneous abortion mouse model. Modulate the balance of Th1/Th2 cytokines will protect fetus from resorption in abortion prone mice (PMID: 29317300; PMID: 8543817.).
Q8: Figure 4C-D: Why are the results presented as folds (normalised with mock) rather than as concentration (pg/ml) if the quantification was performed by ELISA. At which time point post-infection and post-copulation was this assay performed? It could be useful to also include a graph showing the IL12/IL6 ration of mock, delivery, non-pregnancy and miscarriage mice for each group (mock, early pregnancy, middle pregnancy and late pregnancy) as performed in Fig. 2G-I.
Answer:We changed as your suggestion, and shown with both concentration (pg/ml) and IL12/IL6 ratio in the new version.
Round 2
Reviewer 2 Report
The reference by Schlosser et al. still not included, otherwise points addressed.
Author Response
Q: The reference by Schlosser et al. still not included, otherwise points addressed.
Answer: Thank you. We added the reference in Discussion section in the new version.
Reviewer 3 Report
In the present revised manuscript, Yang et al have developed a model of HEV infection in pregnant BALB/c mice. The authors have answered the different points raised. However, the paper still needs additional English proofreading to be suitable for publication.
The following minor comments should also be considered:
- Fig. 2D: Why 2 delivery mice for the serum and 1 for the feces are presented in the graph (middle pregnancy)?
- Fig. F-G: I still think it is not very accurate to make conclusions (that the viral titer in the serum of HEV-infected miscarriage mice are higher than non-pregnant mice and delivery mice) from these two graphs as data from 2 groups infected at different times post-copulation are compiled and compared. This conclusion should be at least moderated in the text.
- Fig-2H: Do you have an hypothesis to explain why RBV has no effect?
Author Response
Reviewer 3
In the present revised manuscript, Yang et al have developed a model of HEV infection in pregnant BALB/c mice. The authors have answered the different points raised. However, the paper still needs additional English proofreading to be suitable for publication.
Answer: Thank you. The new version manuscript has been carefully revised.
The following minor comments should also be considered:
Q1: - Fig. 2D: Why 2 delivery mice for the serum and 1 for the feces are presented in the graph (middle pregnancy)?
Answer: It should be one mouse in the serum, we changed it, thank you.
Q2: - Fig. F-G: I still think it is not very accurate to make conclusions (that the viral titer in the serum of HEV-infected miscarriage mice are higher than non-pregnant mice and delivery mice) from these two graphs as data from 2 groups infected at different times post-copulation are compiled and compared. This conclusion should be at least moderated in the text.
Answer: Thank you. We moderated the tone of comparison in the new version.
Q3:- Fig-2H: Do you have an hypothesis to explain why RBV has no effect?
Answer: KM01 strain is G1634R mutation, which may the reason why RBV has no effect. We added it in the Discussion section in the new version. Thank you for your help.
This manuscript is a resubmission of an earlier submission. The following is a list of the peer review reports and author responses from that submission.
Round 1
Reviewer 1 Report
In the present manuscript, Yang et al have developed a model of HEV infection in pregnant BALB/c mice that could be useful to study HEV pathogenesis during HEV infection. However, this manuscript is very difficult to follow, as the quality of written English used is poor. Some of the data presented and analysis performed should also be reconsidered. Major comments: - I feel it is not accurate to name the different groups: “first, second and third trimester of pregnancy” as it refers to pregnancy in women and not in mice. “Early, middle and late pregnancy” seem more appropriate. - More details need to be provided concerning the SYBR green-based qPCR used in this study. The reference given (14) does not refer to a specific detailed protocol and is not sufficient. Why are mock samples at 102? Cut-off value? - Line 123-126: when were pregnancy-related hormones monitored? Is it possible that the absence of pregnancy observed in 4/9 mice in the “first trimester of pregnancy” group is linked to HEV infection (early miscarriage) rather than absence of pregnancy? - Control: Results from non-pregnant mice are shown in fig. 2 A and B but this group is not described in the method section and figure 1. Are these mice referring to the ones from the “1st, 2nd and 3rd trimester of pregnancy” groups that did not become pregnant? This should be clearly explained within the figure legend. I also found that it would be more pertinent to show the results from these “non-pregnant” mice within the graph from the corresponding group (but in another colour) as the inoculation was performed at different times (similarly to what was done in figure 2I-K). To compare viral titers between “non-pregnant” and “pregnant” mice (and perform statistical analysis), other groups of mice (with no copulation) should have been inoculated at the same time as the “1st, 2nd and 3rd trimester of pregnancy” groups. - Figure 2A-K: It is difficult to analyse the graphs as they are too small and it is difficult to differentiate the different lines/mice within a same graph. - Figure 3C: How was this analysis done (number of mice for each group, time, error bars)? You should mention in the text why you have determined the concentration of IL-12 (Th1 marker) and IL-6 (Th2 marker?). Why IL-6 and not IL-10? - Figure 4A: How was the statistical analysis performed (comparison to which group)? Is it pertinent to perform a statistical analysis here since HEV RNA was quantified at different days post-infection? - Discussion: The “pregnant BALB/c mice” model could be useful to study HEV infection during pregnancy (accessibility, cost, ..) but differences between pregnancy in mice and women exist and should be discussed in the discussion. The fact that high mortality during pregnancy and severe placental disease have been observed mainly with HEV genotype 1 should also be mentioned in the discussion. Minor comments: - Line 33: Which trimester is the author referring to? - Line 133: what do you mean by “HEV infection is more sensitive”?
Author Response
Review1:
Comments and Suggestions for Authors
In the present manuscript, Yang et al have developed a model of HEV infection in pregnant BALB/c mice that could be useful to study HEV pathogenesis during HEV infection. However, this manuscript is very difficult to follow, as the quality of written English used is poor. Some of the data presented and analysis performed should also be reconsidered.
Major comments: Q1:I feel it is not accurate to name the different groups: “first, second and third trimester of pregnancy” as it refers to pregnancy in women and not in mice. “Early, middle and late pregnancy” seem more appropriate.
Answer: Thank you very much for your advice. We had changed “first, second and third trimester of pregnancy” as “early, middle and late pregnancy”.
Q2: More details need to be provided concerning the SYBR green-based qPCR used in this study. The reference given (14) does not refer to a specific detailed protocol and is not sufficient. Why are mock samples at 102? Cut-off value?
Answer: Thank you. An additional reference about the detail protocol of SYBR green-based qPCR used in this study as reference 21. Yes, the value of mock sample is considered as cut-off value.
Q3: Line 123-126: when were pregnancy-related hormones monitored? Is it possible that the absence of pregnancy observed in 4/9 mice in the “first trimester of pregnancy” group is linked to HEV infection (early miscarriage) rather than absence of pregnancy?
Answer: The hormones (estrogen and progesterone) of each matched mouse were monitored every 7 days. We added the estrogen and progesterone in the new version (Fig.2A and 2B). Mouse without elevated estrogen or progesterone level was defined as non-pregnant. Yes, it is possible that early miscarriage occurred within the first 7 days. Moreover, frequently blood collection may cause miscarriage, especially in the early pregnancy. Thus, we first collected blood at 7 days post-copulation. We discussed this question in line 233-239 in the new version.
Q4: Control: Results from non-pregnant mice are shown in fig. 2 A and B but this group is not described in the method section and figure 1. Are these mice referring to the ones from the “1st, 2nd and 3rd trimester of pregnancy” groups that did not become pregnant? This should be clearly explained within the figure legend. I also found that it would be more pertinent to show the results from these “non-pregnant” mice within the graph from the corresponding group (but in another colour) as the inoculation was performed at different times (similarly to what was done in figure 2I-K). To compare viral titers between “non-pregnant” and “pregnant” mice (and perform statistical analysis), other groups of mice (with no copulation) should have been inoculated at the same time as the “1st, 2nd and 3rd trimester of pregnancy” groups.
Answer: Yes, these non-pregnant mice were from “1st, 2nd and 3rd trimester of pregnancy” groups that did not become pregnant. We clarify these non-pregnant mice in Fig.1 and Fig.2 in another (green) colour. According to your suggestion, we combined the results from these non-pregnant mice within the corresponding group with different colour in the new version. Since the concentration of estrogen and progesterone in non-pregnant mice were unchanged during the whole experiment, thus these non-pregnant mice are able to represent the control.
Q5: Figure 2A-K: It is difficult to analyse the graphs as they are too small and it is difficult to differentiate the different lines/mice within a same graph.
Answer: Thank you. The fig. 2 was divided into two figures (fig.2 and fig.3) in the new version, thus, these graphs are clear.
Q6: Figure 3C: How was this analysis done (number of mice for each group, time, error bars)? You should mention in the text why you have determined the concentration of IL-12 (Th1 marker) and IL-6 (Th2 marker?). Why IL-6 and not IL-10?
Answer: The IL6 and IL10 were determined in the end of experiment (delivery or cesarean section). The number for each group was clarified in the fig. 4 in the new version. The error bars were added in the new version. We added the necessary of determination of IL12 and IL6 in methods section in the new version. Yan and his colleagues reported “Th1 cytokines (Interleukin [IL]-2, IL-12, and Interferon [IFN]-γ), and Th2 cytokines (IL-6 and IL-10) in serum and cerebrospinal fluid (CSF) of HIV-negative patients with neurosyphilis before and after treatment, aiming to elucidate roles of CXCL13 and Th1/Th2 cytokines in immune response to…….” (Medicine (Baltimore) 2017 Nov;96(47)); Maher and his colleagues reported “Key immune genes in the Th1 pathway (IFNγ, TNFα), Th2 pathway (IL 10, IL4, IL6) ……” (PLoS ONE 2016;11(10)). According to those references, both IL6 and IL10 are Th2 cytokines. Thus, we determined IL6.
Q7: Figure 4A: How was the statistical analysis performed (comparison to which group)? Is it pertinent to perform a statistical analysis here since HEV RNA was quantified at different days post-infection?
Answer: Thank you. We removed the statistical analysis since HEV RNA was quantified at different days post-infection.
Q8: Discussion: The “pregnant BALB/c mice” model could be useful to study HEV infection during pregnancy (accessibility, cost, ..) but differences between pregnancy in mice and women exist and should be discussed in the discussion. The fact that high mortality during pregnancy and severe placental disease have been observed mainly with HEV genotype 1 should also be mentioned in the discussion.
Answer: Thank you. We had added these in discussion section in the new version.
Q7:Minor comments: - Line 33: Which trimester is the author referring to? - Line 133: what do you mean by “HEV infection is more sensitive”?
Answer: Thank you. It should be higher viral titer. We revised these in the new version.

Reviewer 2 Report
Yang et al. describe the successful infection of Balb/c mice with hepatitis E virus genotype 4 and the influence of the infection on miscarriage and viral titers in the mice.
The fact that they used a high number of mice is much appreciated and the combination of RNA data and immunohistochemistry is good. It is also very good that they covered all three trimesters of pregnancy for their experiments.
However, to improve the impact of the research, additional experiments and adjustments are strongly recommended.
General remarks:
The authors induce infection of BALB/c mice by i.v. injection of suspension of pig feces that had been infected with genotype 4 HEV. In contrast, Li and colleagues (J. Vet. Med. Sci. 70(12): 1359–1362, 2008) were not able to infect C57BL/6 mice with a genotype 4 strain derived from a wild boar when i.v. injected. Schlosser and colleagues (Viruses 2019, 11, 1; doi:10.3390/v11010001) were not able to infect variety of different immunocompromised C57BL/6 strains as well as BALB/c nude mice with a genotype 3 recovered from wild boar.
The authors should therefore include an experiment, where they try to infect their mice with a genotype 3 strain. They also should include the aforementioned papers in their discussion.
Miscarriage, stillbirth and a high mortality in pregnant women are symptoms that are restricted to infection with HEV genotype 1 in humans. The authors need to comment on that fact extensively in the introduction and discussion, since it is one of the main limitations of their model.
The authors did not put their model into a bigger context of animal models or models for infection of the placenta. We think that a recent review from Li et al. (doi: 10.1101/cshperspect.a032581 ) should be cited for animal models; for the human placenta models, Gouilly et al. 2018 ( DOI: 10.1038/s41467-018-07200-2) and Knegendorf 2018 (doi: 10.1002/hep4.1138) should also be included.
There are several grammar and spelling errors. An additional proof-reading is advised.
Figure 2:
They appear to have robust viral titers, especially in the second trimester mice cohort and nice stainings in the IHC. However, they must include an additional cohort treated with Ribavirin to show that it decreases both number of viral copies in feces and in the organs as well as the signal in the IHC.
When mice have miscarriages, are they removed from the cohort in Figure 2 C and G? Because otherwise, their hormonal status might reverse to normal and differences in viral loads could not necessarily be attributed to that.
They also should indicate the titer of stock they use to infect the mice in the materials and Methods section.
The titers look impressive, but they cannot compare the groups very well, because they rarely have common timepoints. They either need to include these, or tone down their statement in line 132-134. In line 127-129 they compare titers in non-pregnant with pregnant mice in feces. However, there is only one data point at day 11, which appears to have died shortly afterwards. This statement therefore has to be removed, because it is impossible to do statistics with only one mouse.
They should include measurements of negative strand RNA in their samples to confirm that replication is occurring.
Figure 3:
The miscarriage results are very interesting for the second trimester mice.
In line 165 it is stated that a Th1 biased immune status is abortion prone. It should be mentioned that the study they cite was done in humans and discussed how suitable they are as a readout for mouse pregnancies.
3B: The authors show that infection of mice with HEV leads to less fetuses. However, infection is after implantation of eggs into placenta. How are there then less fetuses? Were they less from the start, which is unlikely, or did they die in the placenta? Did the authors observe dead fetuses in the mice?
3C: A Th2 biased immune status (line 166) was observed in mice. It is not clear, why this is now considered to contribute to the miscarriages (line 167-168), when it was just mentioned that Th1 biased immune status was accountable for abortion.
In the discussion (line 211-216), it is referred to the experiment in Fig 3 C as having shown a Th1 biased immune status. This needs to be clarified.
Is an IL-12/IL-6 ratio used often in literature to check for either Th1 or Th2 biased immunity?
As the authors hypothesize that miscarriage was linked to immune status in the mice, they should test, if a biased immune status is only true for mice in the second trimester, or if it just more pronounced or does not change at all.
Figure 4:
4A: How did the authors harvest the organs for titer control and how much organ did they use? If they are able to detect 100-1000 copies of viral genomes per gram uterus, does this mean they detect only 1-10 copies per uterus assuming a uterus weight of approx. 10 mg? (Hobson; J. Reprod. Fert. (1983) 68, 457-463)
Similarly, to Figure 2, an additional RBV cohort should be included to show reduced viral load in placenta and uterus.
Discussion
There are more vertical transmission papers in humans than the one cited. If vertical transmission is discussed, it has to be in a broader context. Please expand this section by adding papers on the human situation. There is also literature on vertical transmission in rabbits. Discuss that as well.
Author Response
Review 2:
Comments and Suggestions for Authors
Yang et al. describe the successful infection of BALB/c mice with hepatitis E virus genotype 4 and the influence of the infection on miscarriage and viral titers in the mice. The fact that they used a high number of mice is much appreciated and the combination of RNA data and immunohistochemistry is good. It is also very good that they covered all three trimesters of pregnancy for their experiments. However, to improve the impact of the research, additional experiments and adjustments are strongly recommended.
General remarks:
Q1: The authors induce infection of BALB/c mice by i.v. injection of suspension of pig feces that had been infected with genotype 4 HEV. In contrast, Li and colleagues (J. Vet. Med. Sci. 70(12): 1359–1362, 2008) were not able to infect C57BL/6 mice with a genotype 4 strain derived from a wild boar when i.v. injected. Schlosser and colleagues (Viruses 2019, 11, 1; doi:10.3390/v11010001) were not able to infect variety of different immunocompromised C57BL/6 strains as well as BALB/c nude mice with a genotype 3 recovered from wild boar. The authors should therefore include an experiment, where they try to infect their mice with a genotype 3 strain. They also should include the aforementioned papers in their discussion.
Answer: The infectivity of genotype 4 swine HEV in BALB/c nude mice had been reported in our previous study (BMC Infectious Diseases, 2009), and then we further confirmed that HEV can infect BALB/c regular mice, but failed in C57BL/6 mice (the data had been submitted to another journal, and been suppled for review. We confirmed that genotype 4 HEV is infectious in BALB/c-based nude mice and regular mice, but dull in C57BL/6 mice. Meanwhile, we also confirmed that genotype 3 HEV is not able to infect mice, no matter BALB/c nude mice, BALB/c mice or C57BL/6 mice). Furthermore, Sun and colleagues (Veterinary microbiology 2018, 486 225:48-52) also confirmed that rabbit HEV can infect BALB/c mice. Mongolian gerbil-based HEV model is also susceptive to gt4 swine HEV (Ruiping She et al., J. Viral Hepat. 2017; J. Viral Hepat. 2016; Virus Res. 2016; Virus Res. 2015). In addition, rat HEV infects patient with liver transplantation and causes persistent infection had been also reported (Emerging Infect. Dis. 2018). Recently, Andonov also reported that rat HEV linked to severe acute hepatitis in an immunocompetent patient (J Infect Dis. 2019). Thus, the host rang of HEV is wider, and rodent is also sensitive to HEV infection.
Q2: Miscarriage, stillbirth and a high mortality in pregnant women are symptoms that are restricted to infection with HEV genotype 1 in humans. The authors need to comment on that fact extensively in the introduction and discussion, since it is one of the main limitations of their model.
Answer: Thank you. We added the report about genotype 4 HEV infection during pregnancy leads to preterm birth, premature rupture of membranes and abortion in introduction (line 37-39) and discussion (line 221-224) in the new version.
Q3: The authors did not put their model into a bigger context of animal models or models for infection of the placenta. We think that a recent review from Li et al. (doi: 10.1101/cshperspect.a032581 ) should be cited for animal models; for the human placenta models, Gouilly et al. 2018 ( DOI: 10.1038/s41467-018-07200-2) and Knegendorf 2018 (doi: 10.1002/hep4.1138) should also be included.
Answer: Thank you. We added the information about animal model for HEV, and HEV infection in placenta in the new version.
Q4: There are several grammar and spelling errors. An additional proof-reading is advised.
Answer: We had proof-reading the manuscript by a professional editor.
Q5: Figure 2:
They appear to have robust viral titers, especially in the second trimester mice cohort and nice stainings in the IHC. However, they must include an additional cohort treated with Ribavirin to show that it decreases both number of viral copies in feces and in the organs as well as the signal in the IHC.
Answer: We had treated HEV infected mice with Ribavirin (RBV), Interferon (IFN-α) and Sofosbuvir (SOF). However, treated with single RBV, IFN-α, SOF or combination had no efficient effect on viral clearance. Moreover, RBV fail to clear genotype 4 HEV infection has been reported (Gastroenterology. 2018 Mar;154(4):1199-1201. doi: 10.1053/j.gastro.2017.12.028). More importantly, the teratogenicity of RBV limits its use during pregnancy. The treatment experiment we had submitted to another journal, so we didn’t add in this manuscript.
Q6: When mice have miscarriages, are they removed from the cohort in Figure 2 C and G? Because otherwise, their hormonal status might reverse to normal and differences in viral loads could not necessarily be attributed to that.
Answer: The miscarriage mice in the early, middle and/or late pregnancy were represent in green color in Fig. 2 A-C, and the levels of hormonal (estrogen and progesterone) were showed in Fig. 2 I-J in the new version.
Q7: They also should indicate the titer of stock they use to infect the mice in the materials and Methods section.
Answer: Thank you. We added the titer of stock in the materials and methods section.
Q8: The titers look impressive, but they cannot compare the groups very well, because they rarely have common time points. They either need to include these, or tone down their statement in line 132-134. In line 127-129 they compare titers in non-pregnant with pregnant mice in feces. However, there is only one data point at day 11, which appears to have died shortly afterwards. This statement therefore has to be removed, because it is impossible to do statistics with only one mouse.
Answer: Thank you. We compared the viral titer in serum with same time points at 7 dpi as your suggestion in the new version.
Q9: They should include measurements of negative strand RNA in their samples to confirm that replication is occurring.
Answer: The negative strand RNA was supplied in Table 1 in the new version.
Figure 3:
The miscarriage results are very interesting for the second trimester mice.
Q10: In line 165 it is stated that a Th1 biased immune status is abortion prone. It should be mentioned that the study they cite was done in humans and discussed how suitable they are as a readout for mouse pregnancies.
Answer: We added the information about Th1 biased immune status in discussion section.
Q11: 3B: The authors show that infection of mice with HEV leads to less fetuses. However, infection is after implantation of eggs into placenta. How are there then less fetuses? Were they less from the start, which is unlikely, or did they die in the placenta? Did the authors observe dead fetuses in the mice?
Answer: Yes. Stillbirth was observed in the placenta of mice infected with HEV.
Q12: 3C: A Th2 biased immune status (line 166) was observed in mice. It is not clear, why this is now considered to contribute to the miscarriages (line 167-168), when it was just mentioned that Th1biased immune status was accountable for abortion. In the discussion (line 211-216), it is referred to the experiment in Fig 3 C as having shown a Th1 biased immune status. This needs to be clarified.
Answer: Sorry, this is a mistake. It should be Th1 in line 180, and we changed it in the new version.
Q13: Is an IL-12/IL-6 ratio used often in literature to check for either Th1 or Th2 biased immunity?
Answer: Yes. IL-12/IL-6 ratio is common for Th1 or Th2 biased immunity. For example, Yan Y and his colleagues reported “Th1 cytokines (Interleukin [IL]-2, IL-12, and Interferon [IFN]-γ), and Th2 cytokines (IL-6 and IL-10) in serum and cerebrospinal fluid (CSF) of HIV-negative patients with neurosyphilis before and after treatment, aiming to elucidate roles of CXCL13 and Th1/Th2 cytokines in immune response to…….” (Medicine (Baltimore) 2017 Nov;96(47)); Maher IE and his colleagues reported “Key immune genes in the Th1 pathway (IFNγ, TNFα), Th2 pathway (IL 10, IL4, IL6) ……” (PLoS ONE 2016;11(10))
Q14: As the authors hypothesize that miscarriage was linked to immune status in the mice, they should test, if a biased immune status is only true for mice in the second trimester, or if it just more pronounced or does not change at all.
Answer: Yes. The levels of IL-12 and IL-6 were determined in all mice with or without HEV infection in the same time point. The IL-12/IL-6 ratio was compared with MOCK group (without HEV infection) with statistically significant. Furthermore, Purabi and colleagues had confirmed that a higher Th1 immune status is statistically significant with fetal mortality in FHF, AVH cases, which plays a role in immune alteration in poor maternal and fetal outcomes in HEV infected pregnancy cases (Journal of Hepatology 2011; (54). However, the miscarriage only occurred in middle pregnancy in our study, thus, we added the association of immune status and miscarriage in discussion section in the new version.
Q15: Figure 4:
4A: How did the authors harvest the organs for titer control and how much organ did they use? If they are able to detect 100-1000 copies of viral genomes per gram uterus, does this mean they detect only 1-10 copies per uterus assuming a uterus weight of approx. 10 mg? (Hobson; J. Reprod. Fert. (1983) 68, 457-463)
Answer: The organ of MOCK group were harvested at indicated time in early, middle, and late pregnancy, and all uterus, placenta and liver of fetuses were collected. We rearranged Fig. 4A, which will be clearer.
The copy number of 100-1000 was the cut-off value. The result below the cut-off value was unconvinced.
Q16: Similarly, to Figure 2, an additional RBV cohort should be included to show reduced viral load in placenta and uterus.
Answer: Same answer to Q5.
Q17: Discussion
There are more vertical transmission papers in humans than the one cited. If vertical transmission is discussed, it has to be in a broader context. Please expand this section by adding papers on the human situation. There is also literature on vertical transmission in rabbits. Discuss that as well.
Answer: Thank you. We added more references about vertical transmission in the new version.
Reviewer 3 Report
AUTHORS
Manuscript ID: viruses-432385
Title: Successful establishment of hepatitis E virus infection in pregnant BALB/c mice
The present report aims at clarifying the poor pregnancy outcome of HEV infection using an HEV infected pregnant mice animal model. The paper uses interesting methodological approaches and potentially produce interesting results but some concerns need to be addressed before publishing.
The introduction needs to show the genotypic differences between HEV1/2 and HEV 3/4. Genotypes 3 (HEV3) and 4 (HEV4) are the ones considered emergent, while genotypes 1 (HEV1) and 2 (HEV2) are know to circulate in developing countries for more than half a century. The clinical outcome is also very different and genotypes 3 and 4 do not cause abortions. This needs to be reflected in the introduction.
I believe genotype 4 HEV is not known to cause abortion. Why have authors chosen HEV4 for an abortion model and not the more pathogenic (and well associated to abortions) HEV1? Please consider reading the paper: Gouilly J, Chen Q, Siewiera J, Cartron G, Levy C, Dubois M, Al-Daccak R, Izopet J, Jabrane-Ferrat N, El Costa H. Genotype specific pathogenicity of hepatitis E virus at the human maternal-fetal interface. Nat Commun. 2018 Nov 12;9(1):4748. doi: 10.1038/s41467-018-07200-2. PubMed PMID: 30420629; PubMed Central PMCID: PMC6232144.
Did authors screen the fecal suspensions used for inoculation for other viral agents (authors have used antibiotics and have filtered for bacteria/fungi also)? If not, how can you confirm the clinical outcome of miscarriages is associated to HEV and not other viral agents not screened for?
Author Response
Review 3:
Comments and Suggestions for Authors
AUTHORS
Manuscript ID: viruses-432385
Title: Successful establishment of hepatitis E virus infection in pregnant BALB/c mice
The present report aims at clarifying the poor pregnancy outcome of HEV infection using an HEV infected pregnant mice animal model. The paper uses interesting methodological approaches and potentially produce interesting results but some concerns need to be addressed before publishing.
Q1: The introduction needs to show the genotypic differences between HEV1/2 and HEV 3/4. Genotypes 3 (HEV3) and 4 (HEV4) are the ones considered emergent, while genotypes 1 (HEV1) and 2 (HEV2) are know to circulate in developing countries for more than half a century. The clinical outcome is also very different and genotypes 3 and 4 do not cause abortions. This needs to be reflected in the introduction.
Answer: Thank you. We added the differences of HEV genotypes in introduction section in the new version.
Q2: I believe genotype 4 HEV is not known to cause abortion. Why have authors chosen HEV4 for an abortion model and not the more pathogenic (and well associated to abortions) HEV1? Please consider reading the paper: Gouilly J, Chen Q, Siewiera J, Cartron G, Levy C, Dubois M, Al-Daccak R, Izopet J, Jabrane-Ferrat N, El Costa H. Genotype specific pathogenicity of hepatitis E virus at the human maternal-fetal interface. Nat Commun. 2018 Nov 12;9(1):4748. doi: 10.1038/s41467-018-07200-2. PubMed PMID: 30420629; PubMed Central PMCID: PMC6232144.
Answer: Thank you. The pathogenesis of HEV is largely unclear up to now. Although HEV infection caused adverse pregnancy outcomes were mainly focus on genotype 1. Recently, genotype 4 HEV (4a subgenotype) infection resulted in preterm birth, premature rupture of membranes and abortions also been reported in HEV-infected pregnant women in Qinhuangdao, China (Hepatitis E virus infection and its associated adverse feto-maternal outcomes among pregnantwomen in Qinhuangdao, China.). Meanwhile, as you mentioned the pathogenicity of HEV is genotype specific, thus, we should pay more attention on genotype 4 HEV infection, which is more severe than genotype 3 HEV infection (Genotype 4 hepatitis e virus in france: an autochthonous infection with a more severe presentation).
Q3: Did authors screen the fecal suspensions used for inoculation for other viral agents (authors have used antibiotics and have filtered for bacteria/fungi also)? If not, how can you confirm the clinical outcome of miscarriages is associated to HEV and not other viral agents not screened for?
Answer: The viral agents including the species of enterovirus A and B were exclude by PCR amplified with an universal enteroviruses primers. Meanwhile, porcine reproductive and respiratory syndrome (PRRS) also was excluded by PCR test. Thus, the clinical outcome of miscarriage is caused by HEV infection.